# Optimizing interneuron circuits for compartment-specific feedback inhibition

**Joram Keijser** [1,2]*, **Henning Sprekeler** [1,3]

**1** Modelling of Cognitive Processes, Institute of Software Engineering and Theoretical Computer Science, Technische Universität Berlin, Berlin, Germany, **2** Charité – Universitätsmedizin Berlin, Einstein Center for Neurosciences Berlin, Berlin, Germany, **3** Bernstein Center for Computational Neuroscience Berlin, Berlin, Germany

* keijser@tu-berlin.de

**Data Availability Statement:** All data and code used in this study are available on Zenodo: Keijser & Sprekeler, Code for "Optimizing interneurons for compartment-specific feedback inhibition", Zenodo (2022), https://doi.org/10.5281/zenodo.6320088.

## Abstract

Cortical circuits process information by rich recurrent interactions between excitatory neurons and inhibitory interneurons. One of the prime functions of interneurons is to stabilize the circuit by feedback inhibition, but the level of specificity on which inhibitory feedback operates is not fully resolved. We hypothesized that inhibitory circuits could enable separate feedback control loops for different synaptic input streams, by means of specific feedback inhibition to different neuronal compartments. To investigate this hypothesis, we adopted an optimization approach. Leveraging recent advances in training spiking network models, we optimized the connectivity and short-term plasticity of interneuron circuits for compartment-specific feedback inhibition onto pyramidal neurons. Over the course of the optimization, the interneurons diversified into two classes that resembled parvalbumin (PV) and somatostatin (SST) expressing interneurons. Using simulations and mathematical analyses, we show that the resulting circuit can be understood as a neural decoder that inverts the nonlinear biophysical computations performed within the pyramidal cells. Our model provides a proof of concept for studying structure-function relations in cortical circuits by a combination of gradient-based optimization and biologically plausible phenomenological models.

## Author summary

The brain contains billions of nerve cells—neurons—that can be classified into different types depending on their shape, connectivity and activity. A particularly diverse group of neurons is that of inhibitory neurons, named after their suppressive effect on neural activity. Presumably, their diverse properties allow inhibitory neurons to fulfil different functions, but what these functions are is currently unclear. In this paper, we investigated if a particular function can explain the existence and properties of the two most common inhibitory cell classes: The need to regulate activity in different physical parts (compartments) of the neurons they target. We investigated this function in a computer model, using optimization to find the neuron properties best-suited for compartment-specific inhibition. Our key result is that after the optimization, model neurons largely fell into two classes that resembled the two types of biological neurons. In particular, the optimized

**Funding:** J.K. received a PhD scholarship from the Einstein Center for Neuroscience Berlin (https://www.ecn-berlin.de/). The funders had no role in study design, data collection and analysis, decision to publish, or preparation of the manuscript.

**Competing interests:** The authors have declared that no competing interests exist.

neurons were connected to only one compartment of other neurons. This suggests that the diversity of inhibitory neurons is well suited for compartment-specific inhibition. In the future, our approach of optimizing neural properties might be used to investigate other functions (or dysfunctions) of neuron diversity.

## Introduction

Cortical inhibitory interneurons vary dramatically in shape, gene expression pattern, electrophysiological and synaptic properties and in their downstream targets [1]. Some cell types, e.g., somatostatin (SST)-positive interneurons [2] and some neurogliaform cells in layer 1 [3], predominantly project to pyramidal cell (PC) dendrites. Others—e.g., parvalbumin positive (PV) basket and chandelier cells—primarily inhibit the peri-somatic domain of PCs [4]. Some interneurons receive depressing synapses from PCs, others facilitating synapses [5, 6]. But what is the function of these differences?

One of inhibition's core functions is to prevent run-away excitation [7] by means of feedback inhibition that tracks excitatory inputs. This has led to the concept of excitation-inhibition (E/I) balance [8], i.e., the idea that strong excitatory currents are compensated by inhibitory currents of comparable size. E/I balance is thought to shape cortical dynamics [8, 9] and computations [10, 11] and can be established by means of inhibitory forms of plasticity [12–14]. Selective disruptions of E/I balance are thought to play a key role during learning [15], while chronic disturbances have been implicated with psychiatric diseases, including autism [16, 17] and schizophrenia [18, 19].

Originally conceived as a balance on average [8], E/I balance turned out to be specific to sensory stimuli [20, 21], in time [22, 23], across neurons [24] and to neural activation patterns [25]. The number of excitatory and inhibitory synapses could even be balanced at the subcellular level [26], in a cell-type specific way [27]. Given this high specificity, we hypothesized that excitation and inhibition also balance individually in different neuronal compartments, and that this could be mediated at least in part by compartment-specific feedback inhibition.

Different neuronal compartments often receive input from different sources [28] and integrate these inputs nonlinearly by means of complex cellular dynamics [29, 30]. For example, the apical dendrites of L5 pyramidal cells (PCs) can generate nonlinear calcium events in response to coincident somatic and dendritic inputs [31]. Hence, neuronal output spike trains can have a complex nonlinear dependence on the inputs arriving in different compartments. This poses a challenge for compartment-specific feedback inhibition, which would require interneurons to invert the nonlinear dependence by recovering local dendritic input from pyramidal output. It is therefore far from clear that a compartment-specific feedback inhibition can be achieved at all by means of biologically plausible circuits. If it can, however, it would have to rely on an interneuron circuit that is closely matched to the electrophysiological properties of the cells it inhibits. Parts of the complexity of cortical interneuron circuits could then be interpreted in light of the intrinsic properties of PCs.

Unfortunately, the nature of such a correspondence between the electrophysiology of inhibited cells and suitable interneuron circuits is far from obvious. We reasoned that we could gain insights by means of a model-based optimization approach, in which interneuron circuits are optimized for feedback inhibition onto pyramidal cells with given biophysical properties. Here, we illustrate this ansatz by optimizing interneuron circuits for a nonlinear two-compartment model of L5 pyramidal cells [32]. We show that over the course of the optimization, an initially homogeneous interneuron population diversifies into two classes, which share many

features of cortical PV and SST interneurons. One class primarily inhibits the somatic compartment of the PCs and receives depressing synaptic inputs. The other class primarily inhibits PC dendrites and receives facilitating inputs. We use further computer simulations and mathematical analyses to investigate the mechanism underlying this interneuron diversification. We show how the diversification can be understood from an encoding-decoding perspective, in which the biophysics of the PCs encode two different input streams in a multiplexed code [33], which is in turn decoded by the interneuron circuit. Finally, we identify several factors that determine if interneurons fall into discrete cell types, or exist along a continuum. Together, these findings support the idea that parts of the complexity of cortical interneuron circuits could be interpreted in light of the intrinsic properties of PCs and illustrate how modeling could provide a means of unravelling these interdependencies between the cellular and the circuit level.

## Results

To investigate which aspects of cortical interneuron circuits can be understood from the perspective of compartment-specific inhibition, we studied a spiking network model comprising pyramidal cells (PCs) and interneurons (INs) (see Methods). PCs were described by a two-compartment model consisting of a soma and an apical dendrite. The parameters of this model were previously fitted to capture dendrite-dependent bursting [32]. PCs received time-varying excitatory inputs in both the somatic and the dendritic compartment, and inhibitory inputs from INs. The excitatory inputs to both compartments consisted of alternating currents of varying amplitude. The two currents were initially uncorrelated; we will investigate the effect of this assumption later. INs were described by an integrate-and-fire model. They received excitatory inputs from the PCs, and inhibitory inputs from other INs.

We optimized the interneuron circuit for a compartment-specific feedback inhibition. In the presence of time-varying external input, feedback inhibition tracks excitatory inputs in time [8, 23]. We therefore enforced compartment-specific feedback inhibition by minimizing the mean squared error between excitatory and inhibitory inputs in both compartments, by means of gradient descent with surrogate gradients [34]. Importantly, we optimized not only the strength of all synaptic connections in the network, but also the short-term plasticity of the PC → IN connections (see Methods).

### Interneuron diversity emerges during optimization

Before the optimization, interneurons formed a single, homogeneous group (Fig 1A, top). Most inhibited both somatic and dendritic compartments (Fig 1B, top) and PC → IN connections showed non-specific synaptic dynamics (Fig 1C, top). Synaptic dynamics were quantified using the paired pulse ratio (PPR), the relative amplitude of consecutive postsynaptic potentials (see Methods). A PPR smaller than 1 indicates that later postsynaptic potentials are weaker, and therefore corresponds to short-term depression. A PPR larger than 1 corresponds to short-term facilitation. Excitation and inhibition were poorly correlated, particularly in the dendrite (Pearson correlation coefficients 0.49 (soma) & 0.08 (dendrite)), suggesting that the network did not generate compartment-specific feedback inhibition (Fig 1D, top). The relatively high correlation between somatic excitation and inhibition is explained by the fact that, in a recurrent network, inhibition is bound to track excitation to some extent [8, 23].

During optimization, the interneurons split into two groups (Fig 1A, bottom) with distinct connectivity (Fig 1B, bottom; see also Connectivity among interneurons) and short-term plasticity (Fig 1C, bottom). One group received short-term depressing inputs from PCs and preferentially targeted their somatic compartment, akin to PV interneurons. The other group

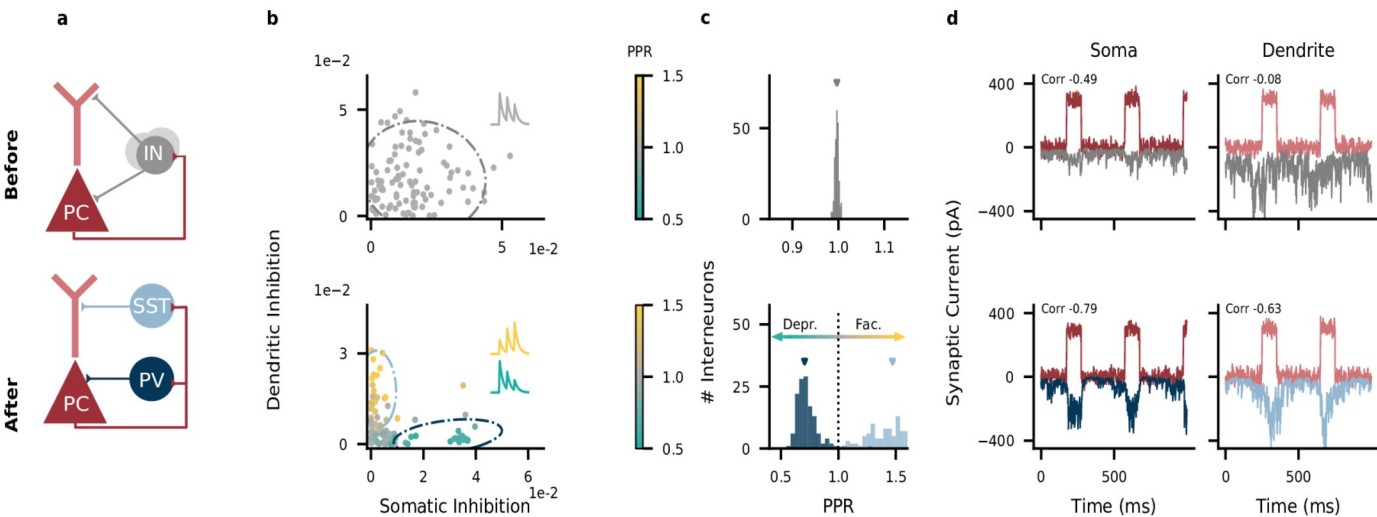

**Fig 1. Interneuron diversity emerges in networks optimized for compartment-specific inhibition.** A: Network structure before (top) and after optimization (bottom). PC, pyramidal cell; IN, interneuron; PV, parvalbumin-positive IN; SST, somatostatin-positive IN. Recurrent inhibitory connections among INs omitted for clarity. B: Strength of somatic and dendritic inhibition from individual INs. Dashed lines: 95% density of a Gaussian distribution (top) and mixture of two Gaussian distributions (bottom) fitted to the connectivity and Paired Pulse Ratio (PPR) data of 5 networks (marginalized over PPR). C: PPR distribution (data from 5 networks). Mean PPR before optimization: 1.00; after optimization: 0.73 (PV cluster, $n$ = 133) and 1.45 (SST cluster, $n$ = 113). D: Excitatory (red) and inhibitory (top: gray, bottom: blue) currents onto PC compartments (average across $N_E$ = 400 PCs). The excitatory inputs to the two compartments are uncorrelated. Inset: correlation between compartment-specific excitation and inhibition.

received short-term facilitating inputs from PCs and targeted their dendritic compartment, akin to SST interneurons. For simplicity, we will henceforth denote the two interneuron groups as PV and SST interneurons. After the optimization, excitation and inhibition were positively correlated in both compartments (Pearson correlation coefficients 0.79 (soma) & 0.63 (dendrite); Fig 1D, bottom). Note that the E/I balance is slightly less tight in time in the dendrites than in the somata (Fig 1D), because synaptic short-term facilitation causes a delay in the signal transmission between PCs and SST interneurons [35, more details below].

To confirm the benefit of two non-overlapping interneuron classes, we performed control simulations in which each interneuron was pre-assigned to target either the soma or the dendrite, while synaptic strengths and short-term plasticity were optimized. Consistent with a benefit of a specialization, the correlation of excitation and inhibition in the two compartments was as high as in fully self-organized networks (Fig 2). Optimized networks with pre-assigned interneuron classes also showed the same diversification in their short-term plasticity, resembling that of PV and SST neurons (Fig 2 and S1 Fig).

## Feedback inhibition decodes compartment-specific inputs

For compartment-specific feedback inhibition, the interneuron circuit has to retrieve the somatic and dendritic input to PCs from the spiking activity of the PCs. This amounts to inverting the nonlinear integration performed in the PCs (Fig 3B). How does the circuit achieve this? Recently, it was proposed that the electrophysiological properties of PCs support a multiplexed neural code that simultaneously represents somatic and dendritic inputs in temporal spike patterns ([33], Fig 3B). In this code, somatic input increases the number of events, where events can either be single spikes or bursts (see Methods). Dendritic input in turn increases the probability that a somatic spike is converted into a burst (burst probability). Providing soma- or dendrite-specific inhibition then amounts to decoding the event rate or burst

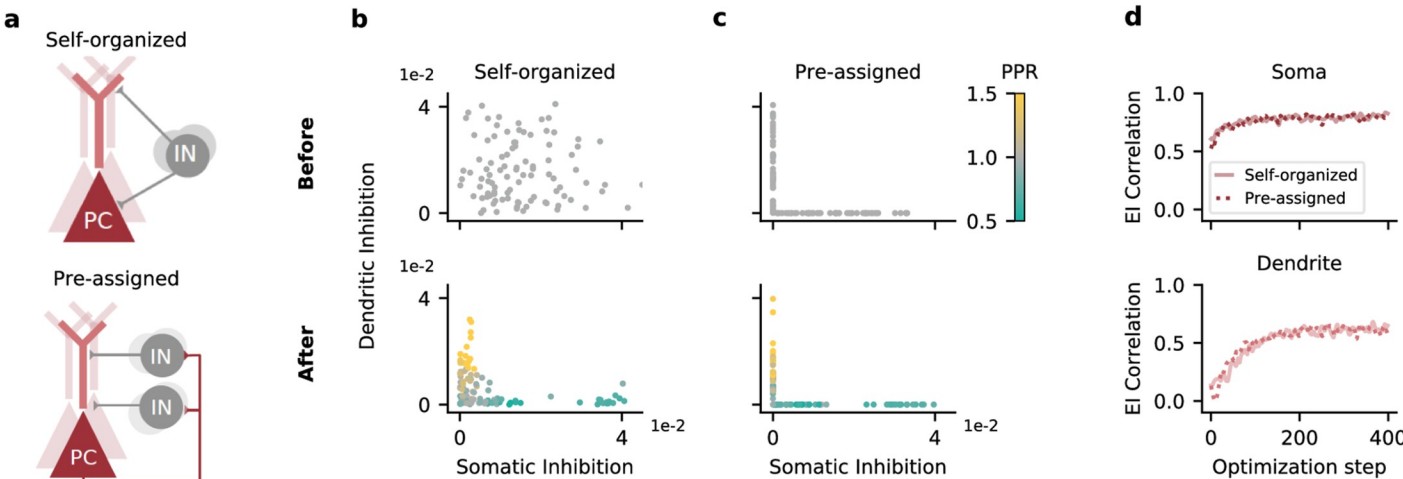

**Fig 2. Compartment-assigned interneurons develop into PV- and SST-like populations.** A: Circuit before learning. Top, interneurons (INs) can inhibit both compartments of principal cells (PCs) and need to self-organize, as in Fig 1. Bottom, INs are pre-assigned to inhibit a single PC compartment. B: IN→PC weights before (top) and after (bottom) optimization. Interneurons self-organize into a population that preferentially inhibits the soma, and a population that preferentially inhibits the dendrites. Data differs from Fig 1 due to random parameter initialization and sampling of training data. C: As B, but with interneurons randomly assigned to inhibit a single compartment (soma or dendrite). Mean PPR: 0.72 (soma-inhibiting population), 1.17 (dendrite-inhibiting population). D: Correlation between compartment-specific excitation and inhibition over the course of the optimization. Solid line: INs were not assigned to a single compartment (Self-organized). Dashed line: INs were assigned to a single compartment (Pre-assigned). Data is smoothed with a Gaussian kernel (width: 2).

probability, respectively. Such a decoding can be achieved in circuits with short-term plasticity and feedforward inhibition [33], and we expected that our network arrived at a similar decoding scheme.

We tested this hypothesis by injecting current pulses to PC somata and dendrites (see Methods). Stronger dendritic input increased the burst probability, which increased the firing rate of SST interneurons via facilitating synapses. The increased SST rate increased dendritic inhibition (Fig 3C–3E, top). Analogously, stronger somatic input increased the event rate, which increased the firing rate of PV interneurons via depressing synapses. The increased PV rate increased somatic inhibition (Fig 3C–3E, bottom). Importantly, inhibition was specific to each compartment (shaded lines indicate input strength to the other compartment): Because PV interneurons were selectively activated by PC events, somatic inhibition was largely unaffected by dendritic excitation. Similarly, SST interneurons were selectively activated by PC bursts, such that dendritic inhibition was largely unaffected by somatic excitation. In the model, interneurons therefore provide compartment-specific inhibition by demultiplexing the neural code used by the PCs.

In networks trained without short-term plasticity, SST neurons were not selectively activated by bursts, and therefore dendritic inhibition did not balance dendritic excitation (Fig 3F and S2 (C) and S3 Figs). A soma-specific E/I balance also required short-term plasticity, but only for weak somatic excitation—consistent with the relatively high somatic E/I correlation at the start of training (Fig 1). In the networks trained without short-term plasticity, the multiplexed neural code was unaltered, because the biophysics of the PCs are the same, but the decoding by the interneuron circuit fails, most prominently for the dendritic input. In our model, short-term plasticity is therefore necessary for compartment-specific feedback inhibition.

## Conditions for the emergence of discrete interneuron classes

PV and SST neurons largely form two non-overlapping cell types [36–38], but in our model they can also exist along a continuum. This depends on three model parameters. First, the

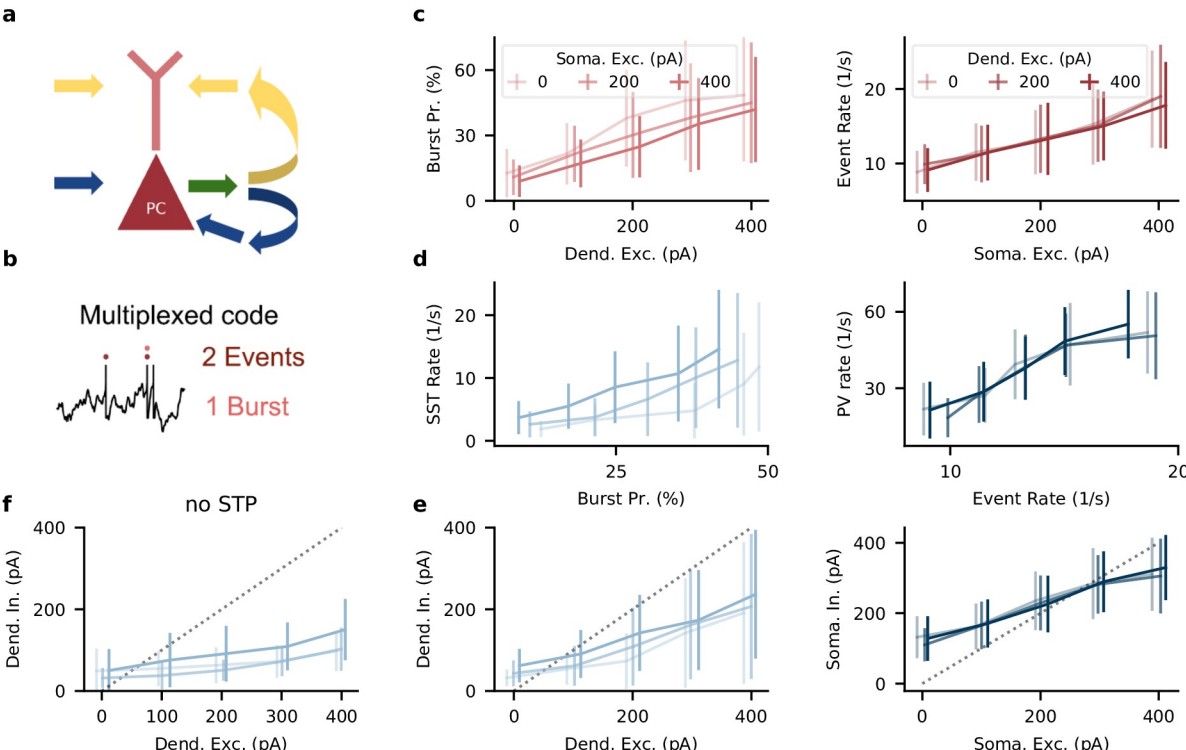

**Fig 3. The interneuron circuit decodes somatic and dendritic inputs to PCs.** A: PC somata and dendrites receive uncorrelated input streams (yellow and blue) that, from PC output spikes (green), have to be separated into compartment-specific inhibition (yellow and blue). B: PCs use a multiplexed neural code. Somatic input leads to events (singlets or bursts). Dendritic input converts singlets into bursts. C, left: Excitatory input to PC dendrites increases burst probability. In this and other top panels (D,E), the shading indicates strength of background somatic input. Right: Excitatory input to PC somata increases event rate. In this and other bottom panels (D, E), the shading indicates strength of background dendritic input. D, left: SST rate increases with bursts probability. Right: PV rate increases with PC events. E, left: dendritic inhibition increases with dendritic excitation, but is only weakly modulated by somatic excitation. Positions on $x$-axis are shifted by 10 pA for visual clarity, error bars indicate sd during 10 stimulus repetitions. Right: somatic inhibition increases with somatic excitation, but is invariant to dendritic excitation. Dashed lines correspond to excitation = inhibition. F: In networks trained without short-term plasticity, dendritic inhibition shows a weaker dependence on dendritic excitation and a stronger dependence on somatic excitation. Also see S3 Fig.

correlations between compartment-specific inputs. So far we assumed that PC somata and dendrites receive uncorrelated input. But recent work suggests that somatic and dendritic activity are correlated [39, 40], potentially reducing the need for compartment-specific inhibition. We therefore tested how correlated inputs affect interneuron specialization by optimizing separate networks for different input correlations. We found that increasing correlation between somatic and dendritic inputs gradually reduced the separation between the interneuron classes (Fig 4A and 4B). For high input correlation, optimized networks contained a continuum in their connectivity and short-term plasticity (Fig 4A and 4B). However, the presence of short-term plasticity was necessary for a dendritic E/I balance for a range of input correlations (Fig 4C). At high correlations, somatic and dendritic inputs are sufficiently similar to make the effect of short-term facilitation negligible. Note that although in this case, distinct interneuron populations were not necessary, the presence of IN classes was also not harmful for E/I balance. A pre-assignment of the interneurons into classes maintained the E/I correlation in both compartments and for any correlation level (S1 Fig).

Interneuron specialization also degraded with increasing baseline activity of the INs (S2 Fig), because high firing rates allow non-specialized inhibition to be canceled by equal and

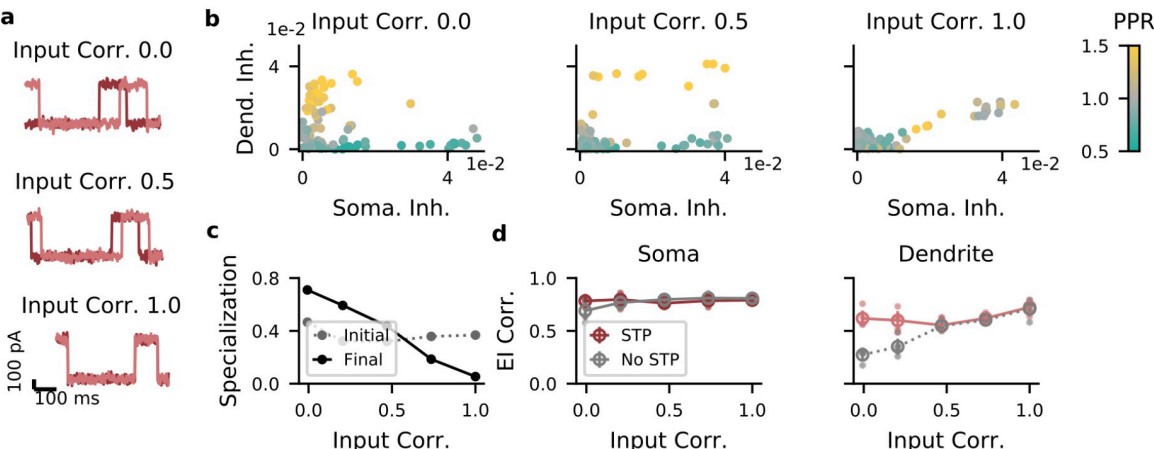

**Fig 4. Correlations between dendritic and somatic input reduce interneuron specialization.** A: Examples for synaptic traces corresponding to different correlation levels. Dark red, somatic current; light red, dendritic input. B: Strength of somatic vs. dendritic inhibition from all INs. Left, middle, right: input correlation coefficient 0 (low), 0.5 (medium), and 1 (high), respectively. C: Specialization of IN → E weights. If each IN targets either soma or dendrites, the specialization is 1 (see Methods). Gray: specialization of initial random network; black: specialization after optimization. D, left: In the soma, excitation and inhibition are balanced across a broad range of input correlations, with or without short-term plasticity (STP). Right: In the dendrites, excitation and inhibition are balanced only with STP when input correlations are small.

opposite disinhibition (see S1 Appendix). The dependence on baseline activity results from the fact that disinhibition is limited by how much the firing of the interneurons can be reduced. In that regard, it is not the baseline firing rate itself that determines the specialization—which is often higher for interneurons than for PCs (see, e.g., [41])—but the relation between the baseline and the dynamic range of the firing rates that is required for the appropriate disinhibition. Note also that a pre-assignment of interneurons into classes again maintained the E/I correlation for different baseline activity levels (S1 Fig).

Finally, interneuron specialization was reduced in networks with heterogeneous inhibitory connectivity. So far, we used homogeneous IN→PC connectivity, i.e., each IN inhibited all PC somata with the same strength, and all PC dendrites with the same strength. In simulations in which interneurons were free to inhibit each soma (and each dendrite) with a unique strength, PV and SST clusters also emerged, but we additionally observed non-specialized interneurons (S4 Fig). But do these non-specialized interneurons play an active role in the computation or are they not necessary and therefore left behind by gradient descent once the problem is solved? The fact that the E/I correlation is not higher than for the homogeneous setting suggests the latter (heterogeneous networks: 0.844 +/- 0.011 (soma) and 0.702 +/- 0.022 (dendrite); homogeneous networks: 0.842 +/- 0.006 (soma) and 0.717 +/- 0.014 (dendrite)).

In sum, a compartment-specific E/I balance seems to require a diversity of interneurons, but the degree to which the interneurons fall into discrete classes depends on a variety of factors.

## Connectivity among interneurons

Because interneurons subtypes also differ in their connectivity to other interneurons [42, 43], we included IN → IN synapses in our optimization. After classifying INs as putative PV and SST neurons using a binary Gaussian mixture model, we found that the connections between the interneuron classes varied systematically in strength. While PV ↔ PV connections, PV →

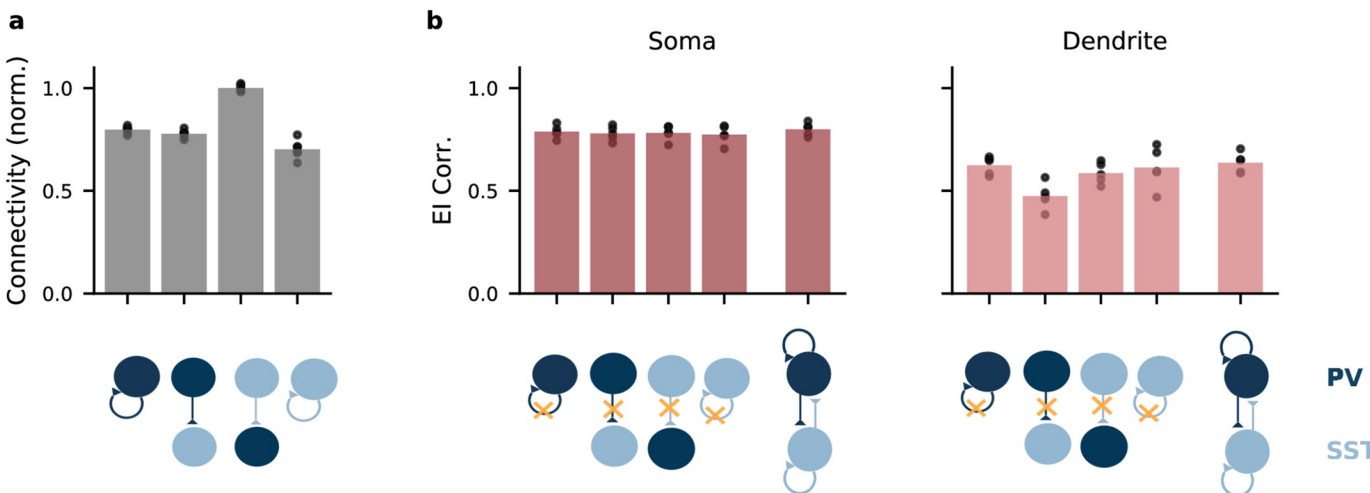

**Fig 5. Recurrent inhibitory connectivity after learning.** A: Connectivity between IN populations. From left to right: PV↔PV, PV→SST, SST→PV, SST↔SST. Bars indicate mean over all networks, dots indicate individual networks. B: Performance as measured by the correlation between excitation and inhibition to PC soma (left) and dendrites (right) of networks optimized while lacking specific connections. Data at the very right: E/I correlation in network with unconstrained connectivity. Only loss of PV → SST connectivity during optimization has a clear effect on dendritic E/I correlations after optimization. Open circles, mean over 5 batches of 8 stimuli with random amplitudes. Small filled circles, individual batches.

SST connections and SST ↔ SST connections were similar in strength on average, SST → PV were consistently stronger (Fig 5A), presumably to compensate for the relatively low SST rates (Fig 3D).

To investigate which connections were necessary, we simulated knockout experiments in networks with pre-assigned interneuron classes, in which we removed individual connections types prior to optimization. We found that only PV → SST connections were necessary for a dendritic E/I balance (Fig 5B). Note that although earlier work did not find PV → SST connectivity in the primary visual cortex of young mice [42], these connections seem to be present in primary visual and somatosensory cortex of older animals [43, 44].

To understand the role of the different IN→IN connections, we performed a mathematical analysis of a simplified network model. The model also contains a population of principal cells (PC) and two populations of interneurons corresponding to PV and SST interneurons, but in contrast to the spiking model, neural activities are represented by continuously-varying rates. The population rates of PV and SST interneurons are denoted by $p$ and $s$, respectively. The activity of PCs is described by two rates: an event rate $e$ that is driven by somatic input and a burst rate $b$ that is driven by dendritic input. The burst rate is assumed to be independent of the somatic input, which is different from BAC firing [31], but generates a linear model that is analytically tractable. The short-term plasticity of a given synapse type is characterized by a single, static parameter, which characterizes the relative efficiency at which events and bursts are transmitted. Synapses for which this parameter is 1 transmit events but not bursts, i.e., they are "perfectly depressing". Synapses for which this parameter is 0 transmit only bursts, i.e., they are "perfectly facilitating". These assumptions allowed us to mathematically analyze the interneuron connectivity required for compartment-specific feedback inhibition. We will only summarize the results, the full analysis is described in S1 Appendix.

Let us first consider the case of dendritic feedback inhibition. The model states that the activity $s$ of the SST neurons is given by a linear combination of the event and burst rate: $s = Ae + Bb$, with factors $A$, $B$ that depend on the connectivity and short-term plasticity in the

circuit in a complicated way. If we assume that SST interneurons target exclusively the PC dendrites, compartment-specific feedback inhibition requires that the activity of SST interneurons depends on dendritic but not somatic input to PCs. Because those two inputs drive the burst rate and event rate, respectively, this condition reduces to the mathematical condition that $A = 0$. Using the dependence of $A$ on the circuit parameters (see S1 Appendix), we get the condition

$$\beta W^{SST \leftarrow PC} - \alpha W^{SST \leftarrow PV} W^{PV \leftarrow PC} = 0 \,, \tag{1}$$

where $W^{Y \leftarrow X}$ denotes the strength of the synaptic connection between population $X$ and $Y$. The two parameters $\alpha$, $\beta$ are the short-term plasticity parameters and quantify how well events are transmitted via the PC→PV and PC→SST connections, respectively.

Condition Eq [1] has an intuitive interpretation. The first term describes how much somatic PC input influences SST activity via the monosynaptic pathway PC → SST. The second term corresponds to the disynaptic pathway PC → PV → SST. The condition therefore states that unless PC→ SST connections are "perfectly facilitating" ($\beta = 0$), the disynaptic PC → PV → SST pathway is necessary (Fig 5) to avoid that somatic input generates dendritic inhibition. The observation that a knock-out of these connections reduces the dendritic E/I correlation in the spiking network (Fig 5B) can therefore be understood as a result of an imperfect facilitation in the PC→SST connection. Indeed, we observed that the synapses in optimized spiking network are not perfectly facilitating. In fact, the Tsodyks-Markram model [45] we used to describe the short-term plasticity in the spiking network cannot achieve perfect facilitation. In the presence of ongoing activity, even for an initial release probability $U = 0$, preceding spikes always leave behind a residual level of synaptic facilitation.

An analogous analysis suggests that disynaptic PC → SST→ PV inhibition is necessary to prevent dendritic inputs from generating somatic inhibition (S1 Appendix), providing a possible function of experimentally observed SST → PV connectivity. At first sight, this appears in conflict with the observation that a knock-out of this connection did not reduce the E/I balance in the soma. However, because bursts are comparatively rare [33], event rate and overall firing (including additional spikes in bursts) are highly correlated. Therefore, the overall firing rate is a good proxy for somatic input and imperfections in synaptic depression in the PC→PV connection do not introduce a sufficiently large problem to necessitate feedforward inhibition via the PC→SST→PV pathway.

## Discussion

Feedback inhibition ensures the stability of cortical circuits [11, 46–48]. Our model indicates that this feedback could operate on a level as fine-grained as different cellular compartments receiving different input streams, and that the required circuitry bears similarity to the one observed in cortex. In particular, we found that an optimization for feedback inhibition led to the emergence of two inhibitory cell classes that resemble PV and SST interneurons in their connectivity and short-term plasticity. This diversification was robust to correlations between somatic and dendritic input, although increasing correlations prompted the SST-like model neurons to contact not only the dendritic, but also the somatic compartment. This is consistent with the extensive branching of cortical SST neurons within the layer that contains their cell body [2]. Even in cases in which the gradient-based optimization did not drive a clear division into cell classes, an artificial pre-assignment of the interneurons did not impair the feedback inhibition.

## Specificity of feedback inhibition

We would like to emphasize that while we optimized for feedback inhibition in different neuronal compartments, the model operates on an ensemble level in the sense that all neurons in the network received the two same time-varying signals in their soma and dendrite. This allows the interneurons to use event or burst rates of the whole ensemble to infer somatic and dendritic inputs with high temporal fidelity [33]. The question of the specificity of feedback inhibition on the population level is an orthogonal one and not fully resolved. The dense and seemingly unspecific connectivity of many interneurons [49, 50] suggests that feedback inhibition operates on the level of the local population, blissfully ignoring the functional identity of the neurons it targets [51]. More recent results have indicated a correlation between the sensory tuning and the synaptic efficacy of interneuron-pyramidal cell connections, however, suggesting that feedback inhibition could operate on the level of functionally identified ensembles [13, 52]. A natural extension of this work would be to endow the pyramidal cells with a tuning to different somatic and dendritic input streams and thereby define functional ensembles. Notably, the ensemble affiliation of a given neuron may differ for soma and dendrite, e.g., two populations of neurons could receive distinct somatic, but identical dendritic inputs. How this would be reflected in the associated feedback-optimized interneuron circuit is an interesting question, but beyond the scope of the present work.

## Origins of interneuron diversity

A natural question for optimization-based approaches is how the optimization can be performed by biologically plausible mechanisms. The gradient-based optimization we performed relies on surrogate gradients [34, 53] and a highly non-local backpropagation of errors both through the network and through time [54, 55], mechanisms that are unlikely implemented verbatim in the circuit [56]. We think of the suggested optimization approach rather as a means to understanding functional relations between different features of neural circuits, i.e., the relation between the biophysics of pyramidal cells and the surrounding interneuron circuits. At this point, we prefer to remain agnostic as to the mechanisms that establish these relations. While an activity-dependent refinement of the circuit is likely, the diversification of the interneurons into PV and SST neurons is clearly not driven by activity-dependent mechanisms alone [57–59]. For example, SST Martinotti cells migrate to the embryonic cortex via the marginal zone, while PV basket cells migrate via the subventricular zone [58]. Their identity is hence determined long before they are integrated into functional circuits. These developmental programs are likely old on evolutionary time scales given that interneuron classes seems more conserved than pyramidal cell classes [60, 61]. We therefore do not expect our mathematical optimization to mimic the evolutionary or developmental processes that generated interneuron diversity.

## Experimental predictions

Given these considerations, we refrain from predictions regarding the optimization process. Still, the model can make predictions regarding the nature of the optimized state. First, it predicts that excitation and inhibition are balanced on short-time scales in both somatic and dendritic compartments. Second, it predicts that PV and SST rates correlate primarily with somatic and dendritic activity, respectively. In our model, this correlation is a consequence of the decoding that underlies the balance (Fig 2), and it is experimentally more accessible than the underlying excitatory and inhibitory currents. Third, inhibiting SST neurons should increase PC bursting, as observed in hippocampus [62] and cortex [63]. The role of short-term facilitation could be tested by silencing the necessary gene Elfn1 [64, 65]. On a higher level, the

model suggests a relation between the biophysical properties of excitatory neurons and the surrounding interneuron circuit. This is consistent, e.g., with the finding that the prevalence of pyramidal cells and dendrite-targeting Martinotti cells seems to be correlated across brain regions [66]. This and other correlations between interneuron and pyramidal properties could be investigated systematically using recent electrophysiological and genomic data from different areas and organisms [67–69].

## Model limitations and extensions

While the synaptic targets and the incoming short-term plasticity of the two emerging interneuron classes are similar to those of PV and SST interneurons, the optimized inhibitory circuitry is not a perfect image of cortex. Aside from the obvious incompleteness in terms of other interneuron types, other features, such as the the often observed weak connectivity from PV to SST neurons [42] did not result from the optimization (Fig 5). Our approach also did not explain further cell type-specific electrophysiological properties: Exploratory simulations indicated that optimizing membrane time constants and spike-rate adaption parameters does not further improve the compartment-specific E/I balance. However, even if our assumption that the interneuron circuit performs compartment-specific feedback inhibition was correct, a perfect match to cortex is probably not to be expected. Firstly, the pyramidal cell model we used is clearly a very reduced depiction of a real pyramidal cell. Because the inhibitory circuitry is optimized for the nonlinear processing performed by these cells, anything that is wrong in the pyramidal cell model will also be wrong in the optimized circuit. It will be interesting to see how the suggested optimization framework generalizes to computations performed by more complex neuronal morphologies [30]. A key challenge in this regard will be the choice of an appropriate computational objective. The objective of E/I balance across different compartments may not generalize to more complex morphologies, because it is unclear if pyramidal cells can multiplex across more than two compartments—because that would require more different spike patterns—let alone that interneuron circuits could invert such a neural code. Secondly, the optimized circuitry is also sensitive to other modelling choices. For example, the circuit separates spikes and bursts by a synergy between short-term plasticity and interneuron connectivity. A wrong short-term plasticity model will therefore lead to a wrong connectivity in the circuit. Here, it will be interesting to see how a more expressive model of short-term plasticity [70] influences the optimal circuit structure. Finally, of course, our optimality assumption could be wrong to different degrees. We could be wrong in detail: Even if the idea of compartment-specific feedback inhibition was correct, our mathematical representation thereof—matching excitation and inhibition in time—could be wrong, with corresponding repercussions in the optimized circuit. Or we could be wrong altogether: PV and SST interneurons serve an altogether different function, and feedback inhibition is merely a means to a completely different end, such as behavioral circuit modulation [63, 71] or the control of plasticity [15, 72]. Here, we did not consider alternative functions for the PV-SST diversity, but this would be an interesting topic for future work.

## Conclusion

Notwithstanding the dependence of the final circuit on specific model choices, we believe that the suggested optimization approach provides a broadly applicable schema for analyses of structure-function relations of interneuron circuits. On a coarser level of biological detail, optimization approaches have recently been quite successful at linking abstract computations to the neural network level [73–75]. While similar in spirit, our approach takes this optimization ansatz from the level of dynamical systems analyses of rate-based recurrent neural networks to

the detailed level of spiking circuits with multi-compartment neurons and short-term plasticity. It will be exciting to see how biological mechanisms on this level of detail support more advanced computations than the mere stabilization of the circuit considered here, but that is clearly a larger research program that extends well beyond the proof of concept presented here.

## Methods

Code and trained models can be found at https://zenodo.org/record/6320089#.Yh4NCyYo-5A. We used Python [76] version 3.7.3, Numpy [77] version 1.18.5, PyTorch [78] version 1.5.1, and scikit-learn [79] version 0.23.2.

### Network model

We simulated a spiking network model consisting of $N_E$ pyramidal cells (PCs) and $N_I$ interneurons (INs), as in earlier work [33]. PCs are described by a two-compartment model [32]. The membrane potential $v^s$ in the somatic compartment is modeled as a leaky integrate-and-fire unit with spike-triggered adaptation:

$$\frac{dv^s}{dt} = -\frac{v^s - E_L}{\tau_s} + \frac{g_s f(v^d) + w^s + I^s}{C_s} \tag{2}$$

$$\frac{dw^s}{dt} = -\frac{w^s}{\tau_{s,w}} + b_s S(t). \tag{3}$$

Here, $E_L$ denotes the resting potential, $\tau_s$ the membrane time constant and $C_s$ the capacitance of the soma. $I^s$ is the external input, and $w^s$ the adaptation variable, which follows leaky dynamics with time constant $\tau_{s,w}$, driven by the spike train $S$ emitted by the soma. $b_s$ controls the strength of the spike-triggered adaptation. $v^d$ is the dendritic membrane potential, the conductance $g_s$ controls how strongly the dendrite drives the soma, and $f$ the nonlinear activation of the dendrite:

$$f(v) = 1/(1 + \exp(-(v - E_d)/D_d)). \tag{4}$$

The half-point $E_d$ and slope $D$ of the transfer function $f$ control the excitability of the dendrite. When the membrane potential reaches the spiking threshold $\vartheta$, it is reset to the resting potential and the PC emits a spike. Every spike is followed by an absolute refractory period of $\tau_r$.

The dynamics of the dendritic compartment are given by:

$$\frac{dv^d}{dt} = -\frac{v^d - E_L}{\tau_d} + \frac{g_d f(v^d) + c_d K(t - \hat{t}) + w^d + I^d}{C_d} \tag{5}$$

$$\frac{dw^d}{dt} = -\frac{w^d}{\tau_{d,w}} + \frac{a_d(v^d - E_L)}{\tau_{d,w}}. \tag{6}$$

In addition to leaky membrane potential dynamics with time constant $\tau_d$, the dendrite shows a voltage-dependent nonlinear activation $f$, the strength of which is controlled by $g_d$. This nonlinearity allows the generation of dendritic plateau potentials ("calcium spikes"). Somatic spikes trigger backpropagating action potentials in the dendrite, modeled in the form of a box-car kernel $K$, which starts 1ms after the spike and lasts 2ms. The amplitude of the backpropagating action potential is controlled by the parameter $c_d$. The dendrite is subject to a voltage-activated adaptation current $w^d$, which limits the duration of the plateau potential. This

adaptation follows leaky dynamics with time constant $\tau_{d,w}$. The strength of the adaptation is given by the parameter $a_d$. Note that the model excludes sub-threshold coupling from the soma to the dendrite.

The interneurons are modeled as leaky integrate-and-fire neurons:

$$\frac{dv^i}{dt} = -\frac{v^i - E_L}{\tau_i} + \frac{I^i}{C_i},\tag{7}$$

with time constant $\tau_i$. Spike threshold, resting and reset potential, and refractory period are the same as for the PCs.

All neurons receive an external background current to ensure uncorrelated activity, which follows Ornstein-Uhlenbeck dynamics

$$\frac{dI^{x,bg}}{dt} = -\frac{I^{x,bg} - \mu_x}{\tau_{bg}} + \sigma_x \varepsilon.\tag{8}$$

Here, $x \in \{s, d, i\}$ refers to the soma, dendrite, or interneuron, respectively, and $\varepsilon$ is standard Gaussian white noise with zero mean and correlation $\langle \varepsilon(t)\varepsilon(t') \rangle = \delta(t - t')$.

In addition, the somatic and dendritic compartments received step currents mimicking external signals (see Network model), as well as recurrent inhibitory inputs. The recurrent input to compartment $x \in \{s, d\}$ of the $i$th principal cell was given by

$$I_i^{x,inh}(t) = -\sum_{j=1}^{N_I} |W_{ij}^{I\to x}| \, s^j(t).\tag{9}$$

where $s^j$ is the synaptic trace that is increased at each presynaptic spike and decays with time constant $\tau_{syn}$ otherwise:

$$\frac{ds}{dt} = -\frac{s}{\tau_{syn}} + S.$$

The compartment-specific inhibitory weight matrices $W^{I\to x}$, $x \in \{s, d\}$ were optimized; the absolute value in Eq 9 ensured positive weights.

The recurrent input to the $i$th interneuron was given by:

$$I_i^{rec} = \sum_{j=1}^{N_E} |W_{ij}^{E\to I}| \, \mu_{ij}(t) \, s^j(t) - \sum_{k=1} |W_{ik}^{I\to I}| s^k(t).\tag{10}$$

The function $\mu_{ij}(t)$ implements short-term plasticity according to the Tsodyks-Markram model [45]. $\mu(t)$ is the product of a utilization variable $u$ and a recovery variable $R$ that obey the dynamics

$$\frac{du}{dt} = -\frac{u - U}{\tau_u} + (1 - u) \cdot F \cdot S,\tag{11}$$

$$\frac{dR}{dt} = -\frac{R - 1}{\tau_R} - u \cdot R \cdot S.\tag{12}$$

$U$ is the initial release probability, which is optimized by gradient descent. $F$ is the facilitation fraction, and $\tau_R$, $\tau_u$ are the time constants of facilitation and depression, respectively. All parameter values are listed in S1 Table.

Finally, the network parameters were scaled so that the membrane voltages ranged between $E_L = 0$ and $\vartheta = 1$. The scaling allowed weights of order $1/\sqrt{N}$, mitigating vanishing or exploding gradients during optimization. All optimization parameters are listed in S2 Table.

## Optimization

We used gradient descent to find weights $W$ and initial release probabilities $U$ that minimize the difference between excitation and inhibition in both compartments:

$$\mathcal{L} = \sum_{t=1}^{T} \sum_{i=1}^{N_E} \left(E_i^s(t) + I_i^s(t)\right)^2 + \left(E_i^d(t) + I_i^d(t)\right)^2. \tag{13}$$

$E_i^x$ and $I_i^x$ are the total excitatory and inhibitory input to compartment $x \in \{s, d\}$ of PC $i$. In most simulations (all except those for S4 Fig), the output synapses from a given neuron to a given compartment type had the same strength, i.e., the optimization of the output synapses is performed for $N_I \times 2$ parameters. For the input synapses onto the INs, weight and initial release probability were optimized independently for all $N_E \times N_I$ synapses.

To achieve small interneuron rates necessary for interneuron specialization (S2 Fig), we subtracted the mean background input from $E_i^x$:

$$I_i^x(t) = E_i^x(t) - \mu_x, \tag{14}$$

such that the interneurons did not fire when their target compartment received its minimum level of external excitation. To propagate gradients through the spiking non-linearity, we replaced its derivative with the derivative of a smooth approximation [34]

$$\sigma(v) = \frac{1}{\left(1 + \beta|v - \vartheta|\right)^2}. \tag{15}$$

We used (surrogate) gradient descent instead of gradient-free methods because of its favourable sample efficiency, and its recent success in optimizing large-scale spiking networks [80, 81]. We used the machine learning framework PyTorch [82] to simulate the differential equations (forward Euler with step size 1 ms), compute the gradients of the objective $\mathcal{L}$ using automatic differentiation, and update the network parameters using Adam [83]. Backpropagation through time requires storing intermediate activation values during the forward pass (network simulation), followed by a backward pass. Our network consist not just of multi-compartmental neurons, but also of a short-term plasticity model that introduces $N^2$ additional variables ($N$ being the network size). Scaling the model to larger network sizes might therefore require approximating gradient descent by a local learning rule, or by gradient-free optimization. The optimized parameters were initialized according to the distributions listed in S2 Table. We simulated the network response to batches of 8 trials of 600 ms, consisting of 100 ms pulses given at 2.5 Hz. The pulse amplitudes were drawn uniformly and independently for soma and dendrites from the set {100, 200, 300, 400}. Training converged within 200 batches (parameter updates). Before each parameter update, the gradient values were clipped between −1 and 1 to mitigate exploding gradients [84]. After each update, the initial release probability was clipped between 0 and 1 to avoid unphysiological values. We trained networks without clipping the gradient or the release probability to confirm that this did not bias the solutions found by the optimization.

## Methods for figures

**Fig 1.** We measured the short-term plasticity of PC → IN synapses by simulating their response to two EPSPs given 10 ms apart, a typical interspike interval within a burst. The paired pulse ratio (PPR) was computed as the ratio of the two excitatory postsynaptic potential (EPSP) amplitudes, such that a PPR > 1 indicates short-term facilitation and a PPR < 1 indicates short-term depression. The PPR of a single IN was defined as the mean PPR of all its excitatory afferents. Clustering of interneurons was done by fitting a single Gaussian (before optimization) or a

mixture of two Gaussians (after optimization) to the three-dimensional distribution of inhibitory weights to the PC soma, to PC dendrites, and the PC→IN Paired Pulse Ratio (PPR). Both models were fitted using Scikit-learn [79] on pooled data from five networks, trained from different random initializations. The density models where fitted on 246 interneurons that were active (firing rate higher than 1 spk/s) and had a medium to strong projection to either soma or dendrites (weight bigger than 0.01). The dashed lines in Fig 1B illustrate the two-dimensional marginal distributions of the somatic and dendritic inhibition. All PCs received the same time-varying input currents, consisting of 100 ms pulses of 300 pA, given at a rate of 2.5 Hz. Correlations between compartment-specific excitation and inhibition were computed between the the currents to the PC compartments, averaged across all PCs in the network.

**Fig 2.** Before optimization, we assigned interneurons to inhibit either PC somata or dendrites by fixing their weights onto the other compartment to zero. Half of the interneurons was assigned to inhibit the soma, the other half was assigned to inhibit the dendrites. Otherwise, weights and initial release probabilities were optimized as before.

**Fig 3.** The definitions of burst rate, burst probability and event rate were taken from Naud & Sprekeler [33]: A burst was defined as multiple spikes occurring within 16 ms. The time of the first spike was taken as the time of the burst. An event was defined as a burst or a single spike. The instantaneous burst rate and event rate were computed by counting the number of bursts and events, respectively, in bins of 1ms and among the population of PCs, and smoothing the result with a Gaussian filter (width: 2ms). The burst probability was defined as

$$\text{Burst Probability} = \frac{\text{Burst Rate}}{\text{Event Rate}} \times 100\%. \tag{16}$$

We injected current pulses of 100 ms duration to either soma or dendrite while injecting a constant current to the other compartment. Currents where varied in amplitude between 100 and 400 pA; the constant current was 0 pA. The figure shows the mean and standard deviation of the total network activity during 10 current pulses. For Fig 3E, we injected simultaneous pulses to the other compartment of amplitude 0, 200 or 400 pA.

**Fig 4.** We varied the correlation between the inputs to soma and dendrites by generating repeating current pulses with different temporal offsets and optimized a network for each offset. The interneuron specialization was defined as

$$\text{specialization} = 1 - \frac{x^T y}{\|x\|\|y\|}, \tag{17}$$

where $x$ and $y$ are $N_I$-dimensional vectors containing the inhibitory weights onto soma and dendrites and $\|\cdot\|$ the $L_2$ norm. If each neuron inhibits either somata or dendrites, but not both, the specialization will be 1. If the weights are perfectly aligned (i.e., interneurons with a strong dendritic projection also have a strong somatic projection), the specialization will be 0. Here and in all figures, the E/I correlation was computed as the correlation between the time series of the compartment-specific excitation and inhibition, after averaging across all PCs. Shown is the mean over 5 batches of 600 ms, where each batch consisted of 8 trials with amplitudes from {100, 200, 300, 400} pA, sampled independently for soma and dendrites.

**Fig 5.** Fig 5A shows the connectivity strength over five networks. We first used the Gaussian mixture models to assign INs to PV or SST clusters, and then computed the mean connectivity between and within clusters for each network. For Fig 5B, we trained networks with predefined interneuron populations to control the interneuron connectivity. Connections between populations were knocked out by fixing them to zero during and after optimization. E/I correlations are computed for 5 batches of 600 ms, where each batch consisted of 8 trials with amplitudes from {100, 200, 300, 400} pA, sampled independently for soma and dendrites.

**S1 Fig.** As for Fig 2, we assigned interneurons to inhibit either PC somata or dendrites. Here, we trained networks for different correlations between compartment-specific external inputs (cf. Fig 4), and baseline activity levels (S2 Fig). We used the 10th percentile as a robust measure of minimum PV rate. The mean and sd PPR of the PV and SST populations computed over all INs that were active (rate larger than 1 spk/s) and provided a medium to strong inhibition to one PC compartment (weight bigger than 0.01).

**S2 Fig.** The minimum rate of PV neurons was controlled indirectly, by varying the baseline inhibitory target current to the soma—A larger baseline requires a higher minimum PV rate. We varied the minimum inhibitory current by subtracting only a fraction $\alpha$ of the baseline excitatory current:

$$I^x(t) = E^x(t) - \alpha \cdot \mu_x, \tag{18}$$

cf. Eq (14). In the simulations, we varied $\alpha$ between 1 and 0.8, leading to a minimum PV rate between 1 spk/s, and 9 spk/s.

**S3 Fig.** We trained network without short-term plasticity by setting the learning rate of the initial release probability $U$ to 0. We ran the optimization for 1600 instead of the usual 400 steps to ensure that any difference between network performance with and without short-term plasticity was not due to slower convergence. We also trained networks using a range of learning rates (between [0.0001, 0.01]) to ensure any difference could not be due to a sub-optimal learning rate. The optimal learning rate for these non-STP simulations turned out to be the same as the default rate for the STP simulations. We trained 5 networks starting from different random weight initializations, and having confirmed that the results were similar across networks, picked one at random to generate the figure.

**S4 Fig.** This figure shows networks in which interneurons project to PC somata and dendrites with a PC-specific weight. That is, the inhibition to PC somata is given by a $N_I \times N_E$ matrix $W^{I \to S}$, and the inhibition to PC dendrites is given by a matrix $W^{I \to D}$ of the same dimensions. In the other simulations, both inhibitory weights were defined by $N_I \times 1$ matrices, such that a particular interneuron projected to all PC somata with the same weight, and to all PC dendrites with the same weight. Clustering of interneurons was done in the two-dimensional weight space, defined by their mean inhibition to soma and mean inhibition to dendrites (averaged over all PCs).

## Supporting information

**S1 Fig. Non-overlapping interneuron populations achieve compartment-specific inhibition for a range of input statistics.** A: Top, performance as measured by compartment-specific correlation between excitation and inhibition of networks trained on different correlations between compartment-specific excitatory inputs. Open circles, mean over 5 batches of 8 stimuli with random amplitudes (see Methods). Small filled circles, individual batches. Here and in the other panels, the interneurons were assigned to inhibit only the soma or only the dendrites. Bottom, interneuron specialization as measured by Paired Pulse Ratio (PPR) decreases with input correlations. Error bars denote sd over IN populations. B: Strength of somatic and dendritic inhibition from individual INs. Top, medium input correlation (0.47); bottom, high input correlation (1.00). Color indicates PPR. C: Top, as A but as function of minimum PV rate. Bottom, interneuron specialization as measured by Paired Pulse Ratio (PPR) is not influenced by minimum PV rate. D: Strength of somatic and dendritic inhibition from individual INs. Top, medium PV rate (4 spk/s); bottom, high PV rate (9 spk/s).
(TIF)

**S2 Fig. Higher baseline PV rates decrease the need for interneuron specialization.** A: Strength of somatic and dendritic inhibition from individual INs. Left, middle, right: network optimized with a baseline PV rate of 1 (low), 5 (medium), and 9 spk/s (high), respectively. B: Specialization of IN→E weights. If each IN targets either soma or dendrites, the specialization is 1 (see Methods). Gray: specialization of initial, random network; black: specialization after optimization. C: Left, correlation between excitation and inhibition as function of minimum PV rate. Red: networks with optimized short-term plasticity. Gray: Networks without short-term plasticity. Open circles, mean over 5 batches of 8 stimuli with random amplitudes. Small filled circles, individual batches.
(TIF)

**S3 Fig. Networks without short-term plasticity fail to achieve a dendrite-specific E/I balance.** A: PCs use a multiplexed neural code both in presence (colors) and absence (gray) of short-term plasticity in their efferents. Top: Excitatory input to PC dendrites increases burst probability. Bottom: Excitatory input to PC somata increases event rate. B, top: SST rate increases with bursts probability only when SSTs receive short-term plastic input. Bottom: PV rate increases with PC events, but absent short-term plasticity only for intermediate and high event rates. C, top: dendrite-specific inhibition requires short-term plasticity. Bottom: soma-specific inhibition requires short-term plasticity only for small somatic input.
(TIF)

**S4 Fig. Networks with heterogeneous IN→PC connections contain PV and SST classes, but also unspecific interneurons.** A: IN→PC weights after optimization, in networks where INs can connect to each PC soma (and dendrite) with a unique strength. Shown are the per-IN weights averaged over all PCs. A Gaussian mixture model identified 4 clusters: a PV and a SST cluster and 2 unspecific clusters. Dots show means, ellipses show 95% density. The PV and SST clusters contain 19% and 15% of the interneurons, respectively. The unspecific clusters with small and large weights contain 52% and 14% of interneurons, respectively. B: PV and SST interneurons receive depressing and facilitating inputs, respectively, as measured by the average paired pulse ratio (PPR), computed over all presynaptic PCs. Arrows indicate means. C: As B, but for interneurons of the two unspecific clusters. Unspecific interneurons do not receive a particular type of short-term plastic input.
(TIF)

**S1 Table. Network parameters.**
(PDF)

**S2 Table. Optimization parameters.**
(PDF)

**S1 Appendix. Mathematical analysis of a simplified network model.**
(PDF)

## Author Contributions

**Conceptualization:** Henning Sprekeler.

**Formal analysis:** Joram Keijser, Henning Sprekeler.

**Funding acquisition:** Joram Keijser, Henning Sprekeler.

**Investigation:** Joram Keijser.

**Methodology:** Joram Keijser.

**Software:** Joram Keijser.

**Supervision:** Henning Sprekeler.

**Writing – original draft:** Joram Keijser, Henning Sprekeler.

**Writing – review & editing:** Henning Sprekeler.

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
