## [Decision Letter · Decision Letter 0]

14 Nov 2021

Dear Mr. Keijser,

Thank you very much for submitting your manuscript "Optimizing interneuron circuits for compartment-specific feedback inhibition" for consideration at PLOS Computational Biology.

As with all papers reviewed by the journal, your manuscript was reviewed by members of the editorial board and by several independent reviewers. In light of the reviews (below this email), we would like to invite the resubmission of a significantly-revised version that takes into account the reviewers' comments.

Both the reviewers agree that the results are interesting and the manuscript is clearly written and well explained. However, there are a number of issues to be clarified about the possible side-effects of the optimisation method chosen and whether the results are robust with respect to model assumptions (e.g. is the interneuron rate too low?)

We cannot make any decision about publication until we have seen the revised manuscript and your response to the reviewers' comments. Your revised manuscript is also likely to be sent to reviewers for further evaluation.

Sincerely,

Abigail Morrison

Associate Editor

PLOS Computational Biology

Lyle Graham

Deputy Editor

PLOS Computational Biology

Reviewer's Responses to Questions

**Comments to the Authors:**

Reviewer #1: The manuscript introduces an interesting approach to study the relationship between connectivity and function in spiking neural network circuits using optimization techniques. This model is used to study how specific connectivity can emerge from functional optimization required by interneurons to decode multiplexed information produced by pyramidal cells and understand the implications of different connectivity patterns in the network.

General questions:

1. Were other fitness rules tested or explored? Could an XOR of the difference in excitation and inhibition between both compartments enhance the speed and performance of the optimization?

2. Which other relevant biophysical components would be interesting to add to this fitness rule in order to enhance the performance of the optimization? Which are the limitations of this rule?

3. Why choose gradient descent? Given that the underlying dynamics are highly non linear, would this affect the shape and smoothness of the parameter space chosen, increasing the difficulty of finding unique minima? Also, the authors slightly hint that this optimization mechanism could be of evolutionary nature.

4. Which limitations are added to the optimization process by the choice that all output synapses from a given neuron to a given compartment type had the same

strength?

5. Which are the foreseen effects in the optimization process derived by the clipping of gradient values before update as well as the clipping of release values after update? Even if not in the scope of this work, it would be interesting to see which parameters does the optimization algorithm find which are not biologically realistic and which meaning they would have.

6. Given that the optimization algorithm benefits from the network processing of input in order to better identify the generated patterns by the pyramidal cells, how to you foresee the size of the network to affect the speed and efficiency of the optimization?

7. The resulting networks depend on several factors mentioned by the authors in the discussion, including plasticity rules and fitness rule chosen. Which framework would the authors suggest to put in place in order to complete a cycle with experimental data which allows to identify if the optimization results are actually correct or not?

8. How would the fitness rule change if the number of compartments change? Do you still think the current approach would hold?

9. The authors mention in the discussion that this optimization approach would be useful to analyze the structural and functional relationships of other spiking neural circuits and also could be extended to cells with more compartments. In order to enable this and for reproducibility of the manuscript, the authors should share their code and data used for the analysis and plots though a public repository. Specific versions of the software used e.g. Tensorflow and Scikit-learn are also missing.

Small typos:

. Equation 13 is missing the subscript i in E^s_i and E^d_i

. In the methods for figures section, PPR and EPSP are used before they are defined.

L452 ...optimized independently...

Reviewer #2: Review is attached as a pdf file.

**Have the authors made all data and (if applicable) computational code underlying the findings in their manuscript fully available?**

Reviewer #1: **No: **The code and the data used to generate the plots has not been made available.

Reviewer #2: Yes

PLOS authors have the option to publish the peer review history of their article (what does this mean?). If published, this will include your full peer review and any attached files.

Reviewer #1: **Yes: **Sandra Diaz Pier

Reviewer #2: No
---

## [Decision Letter · Decision Letter 1]

18 Feb 2022

Dear Mr. Keijser,

We are pleased to inform you that your manuscript 'Optimizing interneuron circuits for compartment-specific feedback inhibition' has been provisionally accepted for publication in PLOS Computational Biology.

Best regards,

Abigail Morrison

Associate Editor

PLOS Computational Biology

Lyle Graham

Deputy Editor

PLOS Computational Biology

Reviewer's Responses to Questions

**Comments to the Authors:**

Reviewer #1: I thank the authors for the revised version of the manuscript. My questions and suggestions have been fully addressed.

Reviewer #2: The authors addressed in a satisfactory way my main concerns. I therefore recommend this manuscript for publication.

One small thing: since the authors admit that STP is necessary for decoding, I would personally prefer it if they removed the sentence "We emphasize that the interneuron circuitry was not optimized to perform this demultiplexing. In particular, our results do not require the interneurons to estimate the event rate or burst probability" (line 144), as it seems to me that demultiplexing was implicitly required. However, I leave that up to the authors.

**Have the authors made all data and (if applicable) computational code underlying the findings in their manuscript fully available?**

Reviewer #1: Yes

Reviewer #2: Yes

PLOS authors have the option to publish the peer review history of their article (what does this mean?). If published, this will include your full peer review and any attached files.

Reviewer #1: No

Reviewer #2: No

---

## [Editor Report · Acceptance letter]

30 Mar 2022

PCOMPBIOL-D-21-01560R1 

Optimizing interneuron circuits for compartment-specific feedback inhibition

Dear Dr Keijser,

I am pleased to inform you that your manuscript has been formally accepted for publication in PLOS Computational Biology. Your manuscript is now with our production department and you will be notified of the publication date in due course.

With kind regards,

Livia Horvath
